# Main Cations and Cellular Biology of Traumatic Spinal Cord Injury

**DOI:** 10.3390/cells11162503

**Published:** 2022-08-11

**Authors:** Constantin Munteanu, Mariana Rotariu, Marius Turnea, Anca Mirela Ionescu, Cristina Popescu, Aura Spinu, Elena Valentina Ionescu, Carmen Oprea, Roxana Elena Țucmeanu, Ligia Gabriela Tătăranu, Sînziana Calina Silișteanu, Gelu Onose

**Affiliations:** 1Faculty of Medical Bioengineering, University of Medicine and Pharmacy “Grigore T. Popa” Iași, 700454 Iași, Romania; 2Neuromuscular Rehabilitation Division, Teaching Emergency Hospital “Bagdasar-Arseni”, 041915 Bucharest, Romania; 3Faculty of Medicine, University of Medicine and Pharmacy “Carol Davila”, 020022 Bucharest, Romania; 4Faculty of Medicine, Ovidius University of Constanta, 900470 Constanta, Romania; 5Balneal and Rehabilitation Sanatorium of Techirghiol, 906100 Techirghiol, Romania; 6Faculty of Medicine and Biological Sciences, “Stefan cel Mare” University of Suceava, 720229 Suceava, Romania

**Keywords:** systematic review, cations, sodium, potassium, lithium, calcium, magnesium, iron, traumatic spinal cord injury

## Abstract

Traumatic spinal cord injury is a life-changing condition with a significant socio-economic impact on patients, their relatives, their caregivers, and even the community. Despite considerable medical advances, there is still a lack of options for the effective treatment of these patients. The major complexity and significant disabling potential of the pathophysiology that spinal cord trauma triggers are the main factors that have led to incremental scientific research on this topic, including trying to describe the molecular and cellular mechanisms that regulate spinal cord repair and regeneration. Scientists have identified various practical approaches to promote cell growth and survival, remyelination, and neuroplasticity in this part of the central nervous system. This review focuses on specific detailed aspects of the involvement of cations in the cell biology of such pathology and on the possibility of repairing damaged spinal cord tissue. In this context, the cellular biology of sodium, potassium, lithium, calcium, and magnesium is essential for understanding the related pathophysiology and also the possibilities to counteract the harmful effects of traumatic events. Lithium, sodium, potassium—monovalent cations—and calcium and magnesium—bivalent cations—can influence many protein–protein interactions, gene transcription, ion channel functions, cellular energy processes—phosphorylation, oxidation—inflammation, etc. For data systematization and synthesis, we used the Preferred Reporting Items for Systematic Reviews and Meta-Analyzes (PRISMA) methodology, trying to make, as far as possible, some order in seeing the “big forest” instead of “trees”. Although we would have expected a large number of articles to address the topic, we were still surprised to find only 51 unique articles after removing duplicates from the 207 articles initially identified. Our article integrates data on many biochemical processes influenced by cations at the molecular level to understand the real possibilities of therapeutic intervention—which must maintain a very narrow balance in cell ion concentrations. Multimolecular, multi-cellular: neuronal cells, glial cells, non-neuronal cells, but also multi-ionic interactions play an important role in the balance between neuro-degenerative pathophysiological processes and the development of effective neuroprotective strategies. This article emphasizes the need for studying cation dynamics as an important future direction.

## 1. Introduction

Spinal cord injury (SCI) is a harsh medical incident that frequently generates severe lifelong disability. SCI affects around 2.5 million people worldwide [1]. Traumatic SCI (tSCI) is a devastating neurological-dysfunctional complex disorder caused by various types of mechanical damaging events, such as falls, aggression/violence, and traffic and/or sports accidents [2].

Most human tSCI lesions are caused by the action of traction and/or compression forces on the spinal cord (usually associated with lesions—mainly fractures—of the spinal column). However, SCI can also happen in non-traumatic circumstances such as tumors, infections, and/or spondylosis, etc., [3]. In addition, the related age distribution shows that males are more prone to experiencing early and late adulthood tSCI, while—somewhat surprisingly—females have higher risks for such pathology during adolescence [4]. Currently, we do not yet have the treatment able to cure this pathology [5]; therefore, its therapy is usually focused on supportive [6], mainly neuroprotective aims, and rehabilitative measures [7,8], with, unfortunately, still minimal results.

The spinal cord comprises various cells and nerve fibers in their specific microenvironment. Some of these include the ascending and descending tracts. As a result of a traumatic event, individuals with spinal cord injuries can experience various degrees of disability [9]. The cellular context of a tSCI immersed in a molecular and ionic microenvironment controls the formation of new synapses, the production of extracellular matrix molecules, the regulation of the maintenance and function of the blood–spinal cord barrier, and modulating actions at the synapses, such as the extracellular absorption of glutamate.

The loss of synaptic connections of neurons with pre- and post-synaptic partners can determine the loss of specific function. The therapeutic strategies need to restore these connections, compromised in primary injury, which are described through the toothpaste tube theory (see Figure 1). After trauma, the morphology and gene expression of astrocytes can be seriously affected, and reactive astrocytes can take on new functional roles in the damaged tissue. Together with other types of cells, such as fibroblasts, can form scar-like structures around the damaged area. This process is known to support axon regrowth following spinal cord injury [10]. Detailed information about SCI morph-path-physiology [11], including biological mechanisms involved in its consequent damages’ evolution and (limited) repair mechanisms, is exhaustively presented in numerous research or review papers.

## 2. Materials and Methods

This systematic review is based on the PRISMA methodology, by searching free full-text available papers written in English, which have appeared in the last five years, using specific keywords combinations (Table 1), in the well-known international databases: Elsevier, PubMed, PMC, ISI-Web of Science and Google. In addition, the inclusion criteria were fixed regarding pathology and interventions. Databases interrogation provided, initially, 207 articles. Appling the eligibility selection filters resulted in 51 unique published qualified studies.

## 3. General Characteristics of the tSCI Cellular Biology

Cells’ external and internal microenvironment [10] involves a cloud distribution of various ions, nutrients [15] (for example, sugars, fats, oxygen), and secreted (neurotransmitters, cytokines, hormones, etc.) and excreted molecules (carbon dioxide, metabolic by-products) in an aqueous medium [4]. Lesioned axons can not regrow/regenerate in the adult mammalian central nervous system (CNS) because of multifactorial endo- and exogenous conditions, including developing a post-injury non-permissive environment [16].

A primary injury is caused by a sudden and unexpected impact that dislocates or fractures vertebrae [17]. Although most tSCIs can completely functionally interrupt the spinal cord, some lesions can still allow a limited spinal cord function at/or below the injured level [18]. In addition, according to clinical observations, the primary injury phase can lead to the destruction of neural parenchyma with cellular deaths and loss of axonal network, associated with hemorrhage [16].

The secondary injuries are usually divided into acute, sub-acute, and chronic phases and are characterized by the loss of neurons and glial cells caused by a very extended and puzzling “cascade of secondary events” [19] calcium influx inside the cells, inflammation, ischemia/hypoxia—this complex reactive process affecting even the differential regulation of numerous genes in the spinal cord tissue [4].

In patients with tSCI, the lack of vascular supply can cause various pathophysiological effects, such as cytotoxic and vasogenic oedemas. The lack of water molecules can also cause cell death and promote cell destruction. In addition, the increasing permeability of the blood-spinal cord barrier can also cause the loss of water and ions from the cells [19]. This effect can then lead to the development of vasogenic edema. So, the acute secondary injuries have, as basic lesions, hemorrhage and inflammation. In this stage, a specific pro-inflammatory protein such as IL-6 can be found in the spinal fluid [20] and thereby may be considered a related structural injury biomarker.

In the sub-acute phase, tSCIs encompass three tissue compartments with distinct cellular composition, represented by a central non-neural scar core, an astroglial scar border, and a surrounding zone demarcated by neural tissue [6]. Cell destruction is caused by various factors such as metabolic disturbances [21], ischemia [22]/hypoxia [23].

Astrocytes and microglia are local cells’ early responders—activated including toxic debris clearance. In addition, they can recruit pro-inflammatory cells by secreting various cytokines and chemokines. Astrocytes growing in the scar border of severely damaged tissue migrate and assemble into a scar network around the core tissue. The formation of this scar border is usually completed in about 2 to 3 weeks after tSCI. The astrocyte scar borders can protect adjacent neural tissue. Unlike glia limitans borders, astrocyte scar borders are smaller and have few cell layers [6].

The central core of the spinal cord lesions is known as the fibrotic scar. It comprises various immune cells and connective tissues, such as macrophages and stromal fibroblasts [24]. About 70% of the cells that populate the CNS are astrocytes, characterized by a star shape and providing support and physiological insulation to neurons. It is widely believed that the scar plays a dual role in developing SCI as both an inhibitory and protective mechanism. However, the complexity of this role has been underestimated. It is difficult to target the scar in therapeutic applications due to its dynamic nature. This is why it is essential to understand the role of the scar in the development of SCI [24].

The damage caused by spinal cord injury can lead to a cascade of bio-molecular events, which starts with tissue bruising or tearing caused by mechanical damage that leads to the disruption of the nerve fibers and blood vessels, deterioration of nerve and glial cells, ischemia, and tissue edema [25]. Following the initial impact, the various bio-molecular events, such as edema, inflammation, neuronal and glial necrosis, or apoptosis can spread beyond the initial site of injury, but ultimately generate a necrotic zone surrounded by glial/fibrotic scarring. The glial scar has the main role to stabilize the spread of secondary injury and acts as a chronic, physical, and chemo-entrapping barrier that, unfortunately, prevents axonal regeneration (Figure 1) [21].

## 4. Calcium and Cellular Biology of SCI

Calcium performs many functions in the body. For example, calcium plays an essential role at the neuromuscular level in controlling excitability and releasing neurotransmitters [26]. Calcium also has an enzymatic and hormonal role (second intracellular messenger), mediating cellular responses to many internal or external stimuli [27]. In addition, calcium has a plastic role in the body through its insoluble combinations (skeleton) and a dynamic role through Ca^2+^ ions [28]. There is a close balance between these two forms of calcium: the decrease in blood Ca^2+^ is compensated by mobilization from the bones, and hypercalcemia produces increased storage in the skeleton, vessels, or increased elimination. An average amount of 9.5–10.5 mg% is found in blood plasma in three forms: (a) protein calcium (Ca proteinate = 4 mg%), a non-diffusible form of calcium; (b) ionized calcium (2 mg%), a diffusible form, biologically active; and (c) non-ionized calcium (crystalloid = 4 mg%), also a diffusible form, represented by some complex combinations (i.e., calcium citrate, chelated complex). There is a constant balance between the fractions of non-diffusible and diffusible forms (ionic, crystalloid). The transition between them is typically achieved quickly, without altering the global level [27].

A significant number of metabolic processes are influenced by small changes in the concentration of extracellular calcium ions. These include excitability of nerve function and neural transmission [29]; secretion of proteins and hormones; coupling between excited cells and effector cells; cell proliferation; blood coagulation; maintaining the stability and permeability of cell membranes; modulation of activity, especially of those enzymes involved in glycogenolysis; mineralization in bone repair [28].

Following spinal cord injury, primary mechanical trauma can cause cell swelling and lysis, increasing the extracellular Ca^2+^ concentration [30]. This phenomenon leads to the development of excitotoxicity (Figure 2). In addition, an increase in extracellular glutamate can also lead to an influx of Ca^2+^ into neurons. The Ca^2+^ pathway is controlled by various subcellular localized channels and pumps proteins. Therefore, these components can also be used to control the dynamics of Ca^2+^ storage [13].

The differential regulation of genes triggers late injury responses. Some of these effects can affect the cell’s metabolism [7]. The increased release of Ca^2+^ and glutamate can also damage the white matter cells and myelin. In addition, the activation of the *N*-methyl-D-aspartate receptor (NMDA) can also lead to the accumulation of Ca^2+^ in the cytoplasm. This can cause the death of specific cells in the first week following an injury. Ca^2+^ overload in the mitochondria can also reduce ATP production and decrease the Na^+^/K^+^ ATPase levels [7]. Ca^2+^ overload in the mitochondria can also cause the production of nitric oxides (NO) and free radicals. It also can lead to mitochondrial permeability transition pores (mPTPs), increasing the water and oxygen influx within a cell’s mitochondrial matrix [32].

An increased concentration of Ca^2+^ can also lead to the activation of the poly ADP ribose polymerase-1 (PARP1) enzyme. This protein is a regulatory enzyme that helps maintain DNA integrity during cell death. The depletion of NAD+ by PARP can cause the failure of glycolysis and cell death. It can also induce the release of a cyclic protein known as AIF, leading to cell death [20]. In addition, Ca^2+^ increases the levels of various calcium-dependent enzymes, stimulating the production of lipases and other enzymes [33].

The calcium signaling pathway is a significant component of the survival of neurons [34]. Conversely, after SCI, the disrupted calcium can lead to the onset of neurodegeneration [35]. Ca^2+^ overload is a significant inducer of axonal injury. Ca^2+^ can enter injured axons through mechanopores or nanopores [36]. This accumulation activates Ca^2+^-dependent enzymes and phosphatases that can cause axonal loss and cytoskeletal disassembly. The upregulation of IP3R is known to cause axonal damage in the spinal cord [37]. Within the first minutes after an injury, a secondary cascade of damage occurs [38]. This stage of injury can last for weeks or months, and its damaging effects are comparable to those of the initial insult. In the first few minutes following spinal cord injury, various processes such as oxidative stress and lipid peroxidation can impair the function of Ca^2+^ pumps and cell membranes. Axons have a Ca^2+^ buffering system that removes the excess Ca^2+^. However, when the amount of Ca^2+^ gets too toxic, this system can fail [38]. The excessive release of ATP after a spinal cord injury can also be triggered by activating the high-affinity purinergic receptors [39]. This signaling system can also contribute to the Ca^2+^ influx [37]. Ca^2+^ can enter the axoplasm through various pathways [26].

Ca^2+^ dysregulation is a key step in this patho-physiological cascade [38]. Unfortunately, the field of Ca^2+^ dynamics, despite developed electrophysiological and imaging techniques, is not well-supported due to the lack of sufficient evidence supporting its harmful effects [33]. The results of Ca^2+^ overload on the cells can have long-term detrimental effects. Even mild Ca^2+^ elevations can increase calcium levels and trigger the production of reactive oxygen species (ROS) [13]. However, it is not well known if the increased Ca^2+^ levels cause cell death or if it results from the depolarized mitochondria [40].

## 5. Sodium and Cellular Biology of SCI

Sodium (Na) is a monovalent cation from the same group as Li, K, Rb, and Cs. Sodium is the most important element of the alkaline metals group. In animals, sodium ions generate an electrostatic charge on cell membranes. This allows the transmission of nerve impulses. Its level is controlled by aldosterone. The Na^+^ balance is maintained by the diffusion of other substances through the membranes [41].

The total quantity of sodium in the body of an adult male is around 92 g, of which half is located in the extracellular fluid at a concentration of 135–145 mmol/L, about ten percent is found in the intracellular fluid at the concentration of ∼10 mmol/L, and the rest is located in the skeleton [42,43]. Different factors, such as metabolic and osmotic factors, interact to maintain the sodium balance [44]. Low sodium levels are uncommon but can usually be caused by certain medications that make people sweat or vomit excessively [45]. The increased calcium and sodium concentration promotes the development of cytotoxic edema and intracellular acidosis. It also triggers the extracellular release of the neurotransmitter glutamate [46].

When glutamate is bound to the kainate or AMPA channels, it can also cause voltage-dependent depolarization. This can increase the cell’s intracellular Na^+^ concentration. Excessive sodium accumulation in the extracellular space can disrupt the cell’s osmotic balance [47]. This can increase the cell’s water consumption. Free radicals [48] can also disrupt the cellular processes by generating ROS. This can then lead to the cell’s programmed death. A high sodium concentration can also lead to the development of edema and intracellular acidosis. It can also cause the excessive accumulation of Ca^2+^ and Na^+^ into the cell. Like the K^+^ channels, the Na^+^ channels are also found within the myocytes and neurons. They are known to generate action potentials and are also affected by SCI [42].

Studies show that Na^+^ plays a secondary role in the cell migration process. Electrochemical gradients of sodium (Na^+^) can drive the transport of ions and sugars in non-excitable cells. These gradients can also be used to move these solutes across plasma membranes. An increase in the sodium in our body can trigger various signaling cascades that regulate post-translational events. If sodium concentration is elevated in our bodies’ fluids and extracellular spaces, our cells have an inward concentration gradient [42]. It generates a membrane potential across plasma membranes and can stimulate long-distance neuronal communication. Besides Na^+^ involvement in the generation of action potentials, it can also trigger a signaling cascade that can elicit a mitogenic response. The human body contains thousands of signaling molecules. This intercellular communication serves as the main component of basic cellular activity.

The Na^+^/K^+^ ATPase is an ion pump that uses the energy produced by the hydrolyzing of an ATP molecule to transport 3 Na^+^ out of a cell and 2 K^+^ into the cell. Aside from being an ion pump, studies have shown that the Na^+^/K^+^ -ATPase can also function as a signaling transducer and a multi-protein scaffold, activating protein kinase signaling [42].

The Na^+^/Ca^2+^ exchanger utilizes the net flux of three or one Ca^2+^ ions to transport the three Na^+^ ions across the cell membrane. The resulting Ca^2+^ extrusion can be initiated in two different directions: forward or reverse.

The Na^+^/H^+^ exchange is involved in the regulation of cell volume. When exposed to hypertonic conditions, cells rapidly lose water and shrink to an equilibrium inside or outside osmolarities [42].

Na^+^ channel blockers can help protect neurons by reducing cellular swelling and enhancing membrane integrity. In addition, they can also prevent cellular death caused by the excessive release of glutamate. Riluzole is a neuroprotective drug that blocks the excessive sodium influx into the central nervous system [49]. It is believed that a sodium channel blockade can preserve the white matter of the spinal cord. It can also prevent the activation of the sodium hydrogen antiporter system and reduce glutamate release. It has been shown to reduce the effects of excitotoxicity and neuromodulation in patients with spinal cord injuries [49].

## 6. Potassium and Cellular Biology of SCI

The human body has about 140 g of potassium, 97% at the intracellular level [50]. The presence of potassium in the cell membrane is known to maintain the normal function of the cell [51]. A tilt in this balance can cause various diseases to develop. Electrolyte balance [52] is crucial for the body’s general functioning [48]. The equilibrium between the body’s electrolyte and water balance is maintained through a series of homeostatic mechanisms. It is characterized by the existence of both intracellular and extracellular fluid [51]. Homeostasis involves the movement of potassium from inside the cells to the extracellular space. The Na^+^/K^+^ ATPase pump helps transport potassium and sodium ions across the cell membrane. The pump maintains the equilibrium between the extracellular fluid and the intracellular fluid through a buffering process that involves the hydrolysis of ATP and generates an electrical gradient [53].

Under normal physiological conditions, axons are wrapped with myelin, except for nodes of Ranvier [54]. The depolarization of the nodal axonal membrane and the presence of adequate sodium ions allow for the generation of action potentials. It is believed that the K^+^ channels located under the myelin act as an efficient axonal conductor. After recognizing the role of K^+^ channels in the pathogenesis of axonal functional loss, potassium channel blockers [36] have been investigated to block these channels and restore axonal conduction in the injured axons [55].

The potassium channels [56] are located on the axons and play a vital role in regulating physiological processes. They are also known to stimulate the production of action potentials. Through demyelination, the K^+^ channels can increase their activity, preventing the generation of action potentials. Demyelination also triggers the activation of the Ca^2+^ exchanger, which can increase the intracellular levels of Ca^2+^ [57].

The Kir4.1 channel [58] is a glial-specific potassium channel in the central nervous system’s astrocytic maintenance. It is believed that this channel helps maintain the extracellular potassium level. This channel has the highest expression in astrocytes. It contributes to the cell’s membrane properties, such as its hyperpolarized resting membrane and low input resistance. Various studies have shown that the downregulation of Kir4.1 can result in multiple physiological and biochemical changes in the central nervous system. A second model of spinal cord injury has also exhibited the effects of Kirnj10 regulation. Studies show that sustained reductions in Kir4.1 protein expression are dependent on the enhancement of DNA methylation. This finding supports the possibility that this protein could be a target for developing effective drugs for treating various disorders [59].

Downregulation of Kir4.1 is common in various central nervous system disorders. In most cases, it occurs in the inner myelin tongue and the perinodal areas of the astrocytes [60]. The loss of Kir4.1 in OPCs and oligodendrocytes can result in various physiological and biochemical changes, such as demyelination and mitochondria damage. The sulfonylurea receptor is a functionalized version of the K^+^ channel that is physically coupled to one of the two members of the K^+^ family [61].

A shift in the balance of H^+^ and potassium occurs when the cell’s serum pH increases. A spike in serum potassium can also be caused by increased plasma osmolality, which can pull in some of the cell’s water [62].

The body can maintain its normal K^+^ balance without altering other metabolic processes. It can also use insulin to push the K^+^ into the cells. Furthermore, carbohydrate metabolism can help maintain the concentration of K^+^ in the body. The ability of a skeletal muscle to donate some of its stored potassium can help restore the K^+^ balance. After experiencing hypoxia, free radicals can trigger the activation of voltage-dependent K^+^ channels in vitro. These findings support the idea that activating these channels is one of the earliest responses to oxidative stress. High-potassium diets can also affect the plasma concentration of K^+^. Vegetables and fruits with high K^+^ content can lead to this effect [62].

Short-term hyperkalemia could be transient or sustained. For example, a person’s excessive potassium intake and decreased excretion through diet or infusion can induce hyperkalemia. Other factors contributing to the high potassium excretion level include diminishing the body’s mineralocorticoid level or activity. The proper distribution of potassium in the body is essential for maintaining cell function. The body has developed various mechanisms to ensure its adequate distribution [62].

The knowledge about how to manage K^+^ imbalance has become more prevalent. Various interventions such as reducing potassium losses and increasing the amount of potassium in the body can be performed to prevent and cure hypokalemia. Balancing the body’s potassium level is a strategy that can be used to treat hyperkalemia. This condition can be treated through medications and procedures designed to reduce its level. In addition, understanding the link between an increase in potassium level and oxidative stress can help identify potential causes of various diseases [62].

In severe spinal cord injury cases, the intraspinal networks may no longer function properly as an integrated part of the spinal cord’s response to the descending inputs. Several small-molecule compounds have been developed to target key regulators, such as ion channels and receptors, and their pharmacological properties have been well characterized [63]. Following spinal cord injury, the pathways that lead to the formation of potassium-chloride cotransporter (KCC2) in the spinal cord can change its expression. The downregulation of KCC2 can contribute to the development of spasticity following spinal cord injury. After the initial 24 h, the damaged spinal cord can start to recover its capacity to generate sustained depolarizations [64].

## 7. Iron and Cellular Biology of SCI

Iron is an essential constituent of life, a metal involved in a wide range of physiological functions, including enzymatic reactions, energy production, oxygen transport, protein synthesis, and DNA repair. It is usually included in a stable form in metalloproteins. Iron can combine with any type of biomolecule and, as such, will adhere to membranes, nucleic acids, proteins, etc. Inorganic iron in redox reactions is also found in iron–sulfur complexes in many enzymes, such as nitrogenase and hydrogenase [65].

Our body generally contains about 4–5 g of iron. Excessive iron is toxic to humans because it reacts with peroxides in the body, producing free radicals. Toxicity occurs when the amount of iron exceeds the transferrin required to bind the free iron. In a complex with protoporphyrin IX, Fe forms the heme, the prosthetic group of proteins: hemoglobin, myoglobin, and cytochromes. Iron is transported by transferrin, a serum glycoprotein that can bind 2 iron atoms and transport them to all tissues. Excess iron is stored in the body by ferritin and is incorporated into apoferritin. Hemosiderin represents histologically amorphous Fe deposits. Iron is an essential component of metabolism due to its property of taking up (passing from ferric to ferrous) or yielding electrons (passing from Fe^3+^ to Fe^2+^) in a relatively straightforward manner [65].

Iron has a functional dualism at the nervous system level, essential for life but toxic at values that exceed the normal range of variation. At the cellular level, iron is needed for cell growth, but in excess, it causes oxidative stress and cell death. In this context, iron levels are well controlled by the mechanisms of iron homeostasis. The primary protection strategy in the central nervous system is the blood–brain/spinal cord barrier [66].

Iron is essential for several basic cellular processes, including mitochondrial ATP generation and DNA replication, so an iron deficiency in the nervous system affects the division of precursor neuronal cells, astrocytes, and oligodendrocytes. Iron is also needed for several specific functions, such as the synthesis of dopaminergic neurotransmitters and the myelination of axons. Anemia in the early stages later causes mental retardation, and iron deficiency in early development causes neurological abnormalities [67].

Accumulation of iron in the neuronal tissue can cause neurodegeneration. Knowing the ability of iron to donate electrons to oxygen, high iron levels can cause the formation of hydroxyl radicals and hydroxyl ions by the Fenton reaction: Fe^2+^ + H_2_O_2_ → Fe^3+^ + OH + OH^−^ [67]. Elevated iron levels can also generate peroxyl/alkoxyl radicals due to Fe^2+^ lipid peroxidation [68]. Spontaneous oxidation of Fe^2+^ can also produce more superoxide radicals. The resulting ROS and RNS can react with various targets. In most cases, apoptosis occurs due to the influx of Ca^2+^, which activates enzymes involved in the breakdown of cellular proteins [4].

Iron chelators, which can reduce iron accumulation and promote functional improvement, theorized that treating patients with spinal cord injuries with iron can improve their recovery [69]. A chelating agent could help minimize iron accumulation in the spinal cord. A different study in mice revealed that a chelating agent could improve locomotor function after a spinal cord injury. As the blood in the spinal cord floods the tissues, iron is released from the carriers, such as ferritin and circulating transferrin. The acidic environment can facilitate the release of free iron from these molecules in the post-injury period. It has also been theorized that macrophages play an essential role in the body’s recovery by taking up and storing iron [70]. Due to the toxicity of iron and its reactivity, treating patients with spinal cord injuries with iron chelators has been considered a promising strategy to protect the tissue. However, although deferasirox effectively reduced systemic iron stores, it did not decrease intraspinal iron levels [70].

It is believed that the easiest way to chelate iron is through circulating iron levels. Since it does not require the presence of deferasirox to cross a cell membrane, it can be quickly done. After spinal cord injury, the proliferation of new oligodendrocytes leads to new cells within the spinal cord [67].

Iron-induced oxidative stress is hazardous because it causes the release of iron from iron-containing proteins, such as ferritin, heme, or Fe-S combinations, forming a positive feedback loop that exacerbates the toxic effect of iron. It is believed that motor neuron death is a process that leads to the atrophy of the primary motor cortex following spinal cord injury. However, the exact mechanisms of this death are still not known. It is believed that excess iron accumulation in the spinal cord can lead to tissue damage and limit the recovery of individuals following a spinal cord injury [71]. Treating this issue with a chelating systemically could help prevent this accumulation [68].

## 8. Zinc and Cellular Biology of SCI

Zinc (Zn) is a trace element that plays a crucial role in the various biochemical pathways of the human body [72]. Zinc has antioxidant properties and is known to reduce the production of pro-inflammatory cytokines. Around 2800 proteins can potentially bind to Zn. In addition, it is known to support the activities and efficiency of the immune system [73]. In humans, the average zinc concentration ranges from 650 to 1100 μg/L [72].

The level of zinc concentration that affects the response of the body’s immune system is known to depend on its availability [74]. As redistributed from serum to the injured site, the high zinc concentration resulting after an injury can help minimize the inflammatory response’s overshooting [75]. It can also contribute to the adequate activation of the downstream target genes [72].

Two main transport proteins are responsible for zinc homeostasis [76]. The zinc transporters ZnT responsible for Zn effluxes and the Zip family are known to increase the cytoplasmic Zn uptake [72]. Due to its rapid distribution into the cellular compartments, zinc is commonly used as a marker for the onset of the inflammatory response following an injury. Therefore, the researchers hypothesized that the zinc concentration in serum could be used to diagnose TSCI based on the patient’s neurological status and the time following an injury [75]. The researchers noted that the sudden decrease in serum concentrations might have been caused by the molecular processes regulating zinc’s movement into the body. It was hypothesized that the activation of specific cellular pathways might have triggered these changes. The activation of the NF-kB transcription factor is known to contribute to the polarization of monocytes. Although the effects of Zn on the polarization of monocytes are not yet apparent, it has been suggested that the M2 phenotype could play a beneficial role in treating TSCI [77].

The reduction in extracellular zinc levels following an injury can increase the frequency of glutamate excitotoxicity [78] and promote the growth of oligodendrocytes [72]. The researchers then explored the effects of supplemental extracellular zinc on the Ca^2+^ currents of mice. They found that the zinc concentration in the urine attenuated the impact of the chemical on the vesicular Zn transport [72]. Recent studies suggest that the disruption of calcium homeostasis and the accumulation of zinc and glutamate could trigger the toxicity caused by zinc-induced neurotoxicity. The importance of zinc distribution has been highlighted in the pathogenesis of diseases. Free zinc within cells has been linked to the development of neurotransmitter functions [75].

Zinc promotes functional recovery in contusion spinal cord injury patients by activating the Nrf2/GPX4 defense pathway. In addition, using zinc can decrease the inflammatory infiltrates in the spinal cord [79].

Treating patients with zinc can improve the morphological changes in their cells during ferroptosis [73]. These changes include the collapse of mitochondria and the rupture of the outer membrane potential. After zinc was administered to mice, the intercellular adhesion factors and inflammatory factors exhibited decreased downregulation. The effects of zinc on astrocytes inhibit the damage caused by spinal cord injury. Furthermore, its impact on the expression of related proteins can help improve wound healing. Due to its anti-inflammatory properties, zinc has been considered a conventional antioxidant. Its clinical application as a treatment for nerve injury has been supported by its effects on immune system regulation [77].

Researchers theorize that the inflammatory-related responses could be used to predict the severity of the acute phase of SCI. They noted that the zinc concentration in the blood is inversely correlated with the severity of the response.

Although zinc can promote motor function in people with spinal cord injury, the exact mechanisms that regulate this recovery are not fully understood. The zinc significantly increased the number of 19 cytokines in the site of an SCI lesion. In addition, the results indicate that using zinc increases the production of G-CSF, which could result in decreased levels of neuronal apoptosis following SCI [80].

Previous studies have shown that zinc can improve the function of neurons and reduce oxidative stress. However, the role of zinc in the metabolism of the neurons after SCI is not yet clear [79].

Restoring the mitochondria function is critical in preventing the onset of ROS effects. This process can help the cells maintain their energy-producing capabilities and avoid developing diseases that damage the mitochondria [79]. Zinc is also a vital element that plays a central role in developing various biological processes. It can also contribute to maintaining the nervous system’s physiological functions.

## 9. Magnesium and Cellular Biology of SCI

Magnesium is an essential trace element in the body that intervenes in many critical physiological reactions (metabolism of carbohydrates, lipids, and proteins, neuromuscular excitability, enzymatic activities, cell permeability, blood clotting, etc.). An adult’s body contains about 25 g of magnesium (Mg). More than half of this amount is found in the bones, a quarter in the muscles, and the rest is distributed mainly in the heart, liver, kidneys, digestive tract, and plasma. The total magnesium content in an average adult is distributed at 1% extracellularly [81], 31% intracellularly, and 63% in the bones [82]. Magnesium deficiency can cause various symptoms. Some of these include hallucinations, paranoia, muscle weakness, and depression. In addition, magnesium levels can be influenced by multiple factors such as hypercalcemia, primary aldosteronism, and alcoholism. In a follow-up study, paraplegic individuals with similar symptoms were observed to have increased urinary magnesium [82].

Free and total magnesium ions play critical roles in various cellular functions. They can trigger different regulatory functions through specific ion channels and enzymes. Recent studies have revealed that membrane significant fluxes of Mg^2+^ can affect the cell’s metabolic cycles and function [83]. In the central nervous system, Mg^2+^ is known to maintain calcium homeostasis. It is also involved in releasing neurotransmitters and in the transmembrane electrolyte flux. Mg^2+^ can stimulate the excitatory neurotransmitters norepinephrine and serotonin, decreasing the NMDA receptors’ action [83].

After spinal cord injury, the recovery of the damaged network requires the remyelination of long tracts. The absence of the architectural framework can impair the formation of new axon growth and free cell migration. The extracellular matrix in the glial scar contains various proteoglycans, restricting axonal regeneration and preventing neurite outgrowth. In addition, these proteoglycans are also known to activate multiple myelin- and neuron-related signals [84]. Following an injury, in the acute phase, which usually lasts for 3 days, the concentration of magnesium drops [85]. The link between the severity of an injury and magnesium concentration has been investigated. It has been hypothesized that this component could play a role in the secondary injury phase. The group severely affected by the condition exhibited increased plasma protein, blood pressure, and erythrocyte albumin. It also revealed an impairment of redox status. A study conducted in 2000 linked the effects of magnesium and oxidative stress on patients with mild to severe injuries. The researchers noted that the plasma ionized magnesium concentration increased 7 days after the injury [83].

An increase in free hemoglobin can be measured in the central nervous system. It can inhibit the Na/K ATPase activity and lead to neuronal depolarization. This impairs the Na/K ATPase activity and leads to secondary calcium influx [86]. The reduction in total tissue of Mg following an injury could be caused by the influences of extracellular Mg^2+^ on the NMDA receptor [87]. Magnesium can also help restore the concentration of adenosine triphosphate after reperfusion and/or ischemia. However, this process can prevent the long-term regeneration of injured networks [57].

Patients’ magnesium level after suffering an injury is inversely predictive of their possible neurological remission. Individuals with low magnesium levels have higher levels of inflammatory cytokines and plasma concentrations of acute-phase proteins. Through its various anti-adrenergic mechanisms, magnesium controls the movement of calcium and inhibits the release of catecholamine from the cell [84]. It also causes a reduction in systemic vascular resistance and helps prevent the accumulation of lactic acid. Contusive spinal cord injuries trigger a pathological cascade that can lead to vasospasm [88].

The onset of acute cell dysfunction and death can be triggered by the destruction of the microvascular supply and the lack of blood flow to the spinal cord. This process involves the formation of aneurysms and microvascular disturbances in the spinal cord. The onset of hemorrhage and progressive edema following an injury can add to the harsh post-injury environment. This process triggers the proliferation of various cell types and the infiltration of pro-inflammatory cytokines [83]. The activation of phagocytes can also contribute to the cell death of injured cells. In addition, excitotoxic injury can also occur by releasing glutamate through the astrocytes [57].

It has been hypothesized that the loss of endothelial cell function could be caused by the accumulation of calcium or lipid peroxidation, which could be reduced through magnesium treatment. The use of magnesium salt to relieve the vascular spasm of central nervous system vessels has been reported in clinical and experimental settings [88].

## 10. Lithium and Cellular Biology of SCI

Many of the proposed lithium treatment cellular action mechanisms suggest that it can inhibit the activities of various signaling pathways. Lithium has been shown to inhibit multiple phosphomonoesterases being structurally similar to magnesium [89]. Lithium can improve the microenvironment in the spinal cord and promote the regeneration and survival of motor neurons [90]. Lithium chloride can boost the secretion of brain-derived neurotrophic factor (BDNF) in the axons of motor neurons following spinal cord injury [91,92]. Lithium chloride can also help restore the function of the blood–spinal cord barrier after a spinal cord injury [93]. LiCl can also inhibit the apoptotic pathway caused by the GSK-3β activity [94]. The potential targets of lithium are known to require catalyzing metal ions. However, lithium is known to inhibit these targets in a non-competitive manner. The concentrations of Li^+^ in the spinal cord and the brain are less than 5 and 2 mmol/L. Lithium levels are influenced by the diet and the medications used for treating diseases. A Na^+^-dependent countertransport system regulates the effects of lithium on the plasma membrane. This mechanism involves the stimulation of the Li^+^ uptake by external Na^+^ and reducing the Li^+^ uptake by internal Na^+^.

Ion-gated channels are also involved in the distribution of lithium across a cell membrane. The presence of these channels helps regulate the steady-state concentrations of lithium in the cytoplasm. The Na^+^/H^+^ exchanger is a ubiquitous protein that is commonly found in cells. It carries lithium ions in place of sodium. The Na^+^/K^+^ –ATPase pump is also involved in the excitability of neurons. The lithium concentration in the cell does not affect the membrane resistance of neurons. It also triggers a decrease in the free potassium concentration. Lithium can also be substituted for potassium or sodium on specific transport proteins, usually carrying potassium or sodium. These proteins can provide a pathway for lithium to enter a cell. The main entrance ways to lithium are through the sodium channels and the Na^+^/H^+^ exchangers. These channels allow lithium ions to enter the cell without energy consumption [95].

Lithium can also affect the uptake of K^+^ into astrocytes by blocking the reuptake of the K^+^. It has also been observed that its interaction with the electrogenic Na^+^/K^+^ pump can trigger membrane hyperpolarization. An increase in the concentration of Li^+^ can also cause voltage changes in the microelectrodes sensitive to K^+^. It is believed that membrane hyperpolarization is caused by the activation of the Na^+^/K^+^ pump, 15 mmol/L LiCl can reduce or even abolish this hyperpolarization [96].

The effects of lithium on inositol metabolism have been studied. Both inositol and lipids play a complex role in regulating cellular functions [97]. Activating the catenin signaling pathway promotes cell proliferation in non-neural cells. Lithium also partially blocked the Golgi apparatus fragmentation. The disassembly of the Golgi complex is initiated by activating the GSK-3 signaling pathway. GSK-3 has been known to contribute to the pro-apoptotic signaling activity of cells. The activation of the phosphatidylinositol-3 kinase/Akt-signaling pathway can protect cells from pro-apoptotic stimuli [98].

The regulation of cell survival is a crucial component of normal physiology. It can lead to the accumulation of excess or insufficient cell death, which can result in pathological conditions. Neurotrophic factors and the phosphatidylinositol 3-kinase/Akt pathway can also promote cell survival by blocking the proliferation of apoptosis. The Akt protein is a multi-isoform serine/threonine kinase downstream of the PI 3-K protein. The upstream phosphorylation of Akt initiates its activation by PI-3-dependent kinases. Both glutamate and low potassium-containing cultures can induce apoptosis. It has been observed that lithium can also reverse the effects of these two modes of cell death. A treatment with Li^+^ can prevent the induction of low-K+ -induced apoptosis [99].

Lithium’s effects on the NGF pathway can be evidenced by rapid cell surface changes and increased protein phosphorylation. Lithium ions can also affect the responses of cells to polypeptide signals. It can increase insulin and epidermal growth factors and inhibit the activation of NGF-dependent signaling [100].

Studies on the effects of lithium ions on the metabolism and synthesis of neurotransmitters have yielded inconsistent results. However, the functional connections between regulatory genes and cis-regulatory sequences have been suggested as critical factors that influence the response of organisms to lithium [95].

The effects of Li^+^ on the central nervous system are caused by its interventions in potassium homeostasis. This impairs the ability of the isolated spinal cord to generate spontaneous synaptic activity. It has been hypothesized that Li^+^ could compete with K^+^ for the binding site of an extracellular binding site. The presence of high concentrations of Li^+^ in the cell can decrease the free Na^+^ concentration and reduce the stimulation of the Na^+^/K^+^ pump. After exposing the cells to concentrations of Li^+^, the time course of the depolarization is consistent with the effect of the Li^+^ on the membrane [95].

Li^+^ is known to exert several beneficial effects on axonal regeneration and also has remyelinating effects. Therefore, this molecule could constitute a potential therapeutic remedy for nerve injuries in which axonal lesions and demyelination coexist. Furthermore, there is evidence that the GSK3β could be considered an essential factor in the expression of myelin genes, and they open approaches to treating nerve injuries that use inhibitors of GSK3β such as lithium [101].

## 11. Discussion

Aside from the endogenous “braking machinery” for regrowth/regeneration afore enumerated, the complex spinal cord microenvironment is one of the main factors that might influence the regeneration and functional recovery of the respective injured nervous tissue. The post-lesional imbalance at the gene-/molecular, cellular, and tissue levels expresses the disturbance of various inhibitory and/or stimulatory physiological and pathological mechanisms. Reversing the perspective, i.e., from the tissue to the molecular point, Fan Baoyou et al., 2018 [102] emphasize that hemorrhage and ischemia/hypoxia can produce drivers of the microenvironment alterations, expressed mainly by demyelination and glial scars (tissue level); differentiation of stem cells, the transformation of the microglia and oligodendrocyte phenotypes, infiltration of macrophages, and activation of astrocytes represent the cellular level of the tSCI damages, and involvement of chemokines, cytokines, (other) pro-inflammatory and, on the other hand, neurotrophic factors correspond to the molecular level of imbalances. To all these above-presented data [102], we consider in our review a distinct level of microenvironment imbalances, the ionic level, as presented in detail in the previous sections.

The primary mechanical damage can disrupt the blood supply to the spinal cord and the topical capillaries. The lack of blood supply but also bleeding into the spinal cord parenchyma can increase the release of specific ions and macromolecules from the cells, likewise edema which enhances the pressure in the surrounding vessels and causes, in a vicious cycle, ischemia. The lack of ATP can cause an ion imbalance, too, and, again, in a vicious cycle, this could also worsen the neural tissue edema caused by water accumulation in the cells [102]. Unfortunately, remyelination can also be delayed or obstructed by cellular debris in the microenvironment. The accumulation of this debris can limit the extent and quality of the process [102].

Ion imbalance is a known factor that can regulate the pathological changes caused by spinal cord injury. After tSCI, the K^+^, Na^+^, and Ca^2+^ channels are altered. Following tSCI, the levels of Ca^2+^ and Na^+^ are upregulated in the cell, while the concentrations of K^+^ and Mg^2+^ are upregulated extracellularly. The influx of Na^+^ into the cell leads to cytotoxic cellular edema, with water gradually accumulating. On the other hand, previous studies had shown that blocking the Na^+^ channel could also lead to detrimental impacts such as the activation of intracellular phospholipases and intracellular acidosis [103].

Ca^2+^ is a vital component of the CNS that participates in various physiological and pathological processes. As previously pointed out, following spinal cord injury, the concentration of Ca^2+^ increases rapidly. High Ca^2+^ levels can cause apoptosis and/or necrosis by increasing the production of free radicals. It can also damage the white matter [104].

The disruption of the myelin of axons’ sheath following tSCI can lead to different detrimental effects, too. The K^+^ channel’s increased activity can cause demyelination, too. Although the voltage-gated K^+^ channels are essential for remyelination, they can also be affected by 4-aminopyridine (4-AP), a K^+^ channel antagonist [61].

Iron is also a vital component of normal CNS functioning. It can be increased by the influx of red blood cells consecutive to bleeding in the spinal cord. In addition, treating patients with tSCI with deferoxamine can decrease the production of free radicals and promote the survival of the cells. It has also been reported that it prevents the formation of glial scar and reduces the iron ion level [70].

Biomarkers are objectively quantifiable biological characteristics and represent a very attractive and unbiased tool for assessing SCI severity. Furthermore, markers associated with the pathophysiology of acute SCI should aid in monitoring the biological effects of a candidate treatment and also may identify potential targets for novel therapies that reduce secondary damage [105]. A major focus of SCI research is on evaluating the ability of agents to improve recovery after injury. The drug discovery field has adopted the term “perturbagen” to refer to small molecules, peptides, antibodies, oligonucleotides, and so on that, alter a biological process by interfering with one or more molecular targets [106]. To this paradigm, we need to add ions dynamics at the injury site.

## 12. Conclusions

The concept of the microenvironment imbalance after spinal cord injury explains the various alterations and disturbances at the gene/molecular, subcellular, and tissue levels following tSCI. An interesting parallel might be imagined between the microenvironment of an injured spinal cord and the macroenvironment in which we live, represented by air containing oxygen, carbon dioxide, and different aero-ions. The post-tSCI imbalances at the microenvironment level are like “a storm in our climatic ambient” with considerable damaging force. The universal presence of ions, with their chemical and physical properties, has influenced the evolution of cellular mechanisms and further of living beings. Life started through the capacity of biomolecules to manipulate ions and generate “the cellular weather”. Influencing this “weather” can be a desirable intervention in neuro-regeneration. Starting from this paradigm, detailed research on cations dynamics involved in various regeneration mechanisms after SCI is critical in future directions, which might be more feasible in light of the recent spread of biosensors for detecting individual cations.

## Figures and Tables

**Figure 1 cells-11-02503-f001:**
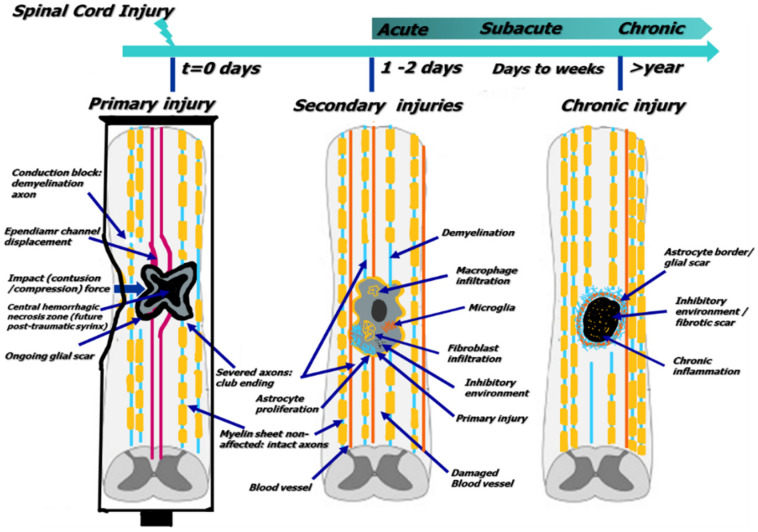
General characteristics of the cellular biology of SCI (preparation based on [12,13,14]).

**Figure 2 cells-11-02503-f002:**
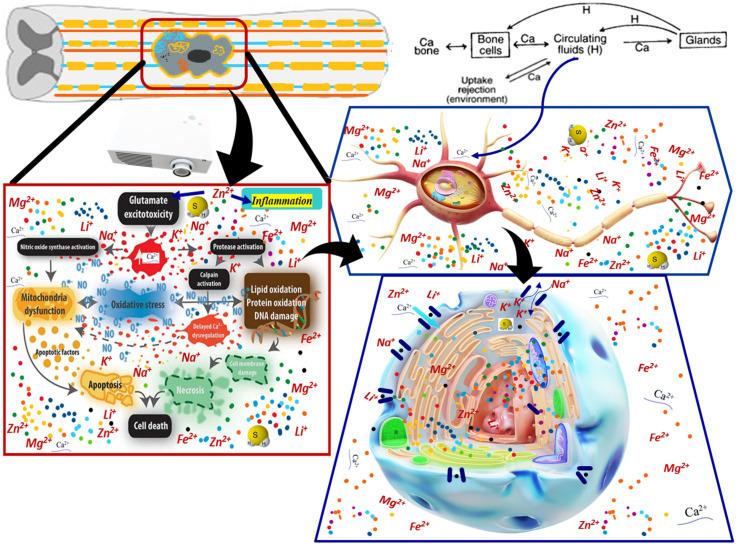
Ionic microenvironment and bio-molecular influenced processes in tSCI. There are presented successive “zoom images” (as follows the black arrows) starting from the tissular view of the injury to the intracellular level of the ionic microenvironment, pointing to the main biological processes influenced by the ions dynamics. The Ca^2+^ pathway is controlled by various subcellular localized channels and pumps proteins. Therefore, these components can also be used to control the dynamics of Ca^2+^ storage (figure preparation is based on [31]).

**Table 1 cells-11-02503-t001:** The keywords combinations used for the contextual search in international databases.

Specific Keywords in Title or Abstract	Elsevier	PubMed	PMC	ISI	Google	Total
“Calcium” AND “Spinal Cord Injury”	21	29	19	6	4	79
“Sodium” AND “Spinal Cord Injury”	9	13	7	8	3	40
“Potassium” AND “Spinal Cord Injury”	4	7	2	3	11	27
“Lithium” AND “Spinal Cord Injury”	5	2	9	7	5	28
“Magnesium” AND “Spinal Cord Injury”	2	3	2	1	7	15
“Iron” AND “Spinal Cord Injury”	0	0	6	0	0	6
“Zinc” AND “Spinal Cord Injury”	0	0	8	0	4	12
Total	41	54	53	25	41	207

## Data Availability

Not applicable.

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
