# Peer review of "Main Cations and Cellular Biology of Traumatic Spinal Cord Injury"

_cells, 2022, doi:10.3390/cells11162503_

Round 1

Reviewer 1 Report

 In this systematic review, the authors have described compilated data about the influence of cations after spinal cord injury, such as cations dynamics after injury, and the possibility of therapeutic targeting. Although the general idea is very interesting, the main text can be improved.

Bellow some comments:

1)   Figure 1: the arrows can be replaced by a figure legend showing the different symbols used in the figures, and what each one is representing. Figure caption also can be improved, by adding a detailed description of each of the depicted phases.

2)   Standardize the use of superscript text, for example, Ca²+, Na+, etc.

3)   Frequently, one paragraph is comprised of several short sentences, usually from the same reference. In my point of view, this makes the reading difficult. Rewriting the paragraphs, and remodeling them to a more fluid text, can greatly improve the review.

4)   Screen the text for some missing references. Some sentences appear to have the reference missing. For example, on page 8 lines 317 to 321.

5)   Authors must upgrade the topics by highlighting the influence of the cations after spinal cord lesions and/or in the repairing process.

Author Response

Thank you very much for your evaluation and detailed analysis of our submitted manuscript. Your comments and advice helped us to improve it, and we consider your amendments very constructive.  Point by point, our responses can be found below:

1)   It was added a new figure to explain the concept of ionic microenvironment better

2)   It has been checked the entire text regarding the use of superscript text for ions.

3)   The entire text was checked. The paragraphs were rewritten to a more fluid text.

4)   The entire text was checked for some missing references. We added the needed references in our revised version.

5)   We upgraded the topics by highlighting the influence of the cations after spinal cord lesions and/or in the repairing process.

Reviewer 2 Report

While this is an extremely interesting topic, this paper fell short of expectations. The writing was at times too 'conversational' and the conclusions were vague. Accordingly, the overall impact of the paper was greatly reduced.

The one and only figure was not at all helpful, low resolution, and poorly made. There are a vast number of similar figures in prior papers, so what was the point of this figure. Something more directly aligned with the message of the paper would be more fitting, if made with professional software. 

Author Response

Dear Reviewer,

Thank you very much for considering our topic extremely interesting.

We tried to increase the impact of our manuscript and to respond to a high level of expectations.

Point by point, our responses are the following:

-    In the revised version of our manuscript were eliminated the too 'conversational' paragraphs.
-    The conclusions were updated.
-    The images of the article were improved significantly, pointing better the concept of the ionic microenvironment.

Reviewer 3 Report

In this review manuscript, the authors did a good job in summarizing cation dynamics and their relevance to spinal cord injury (SCI). This is an important topic yet there is limited knowledge about this. Understanding this may provide direct insights into developing therapeutic strategies. For several important cations, the authors review the general regulatory mechanisms in physiological conditions. In general, the description is pretty accurate and comprehensive. The summary of relevant work in the context of SCI is relatively weak. The authors should point this out as a major future direction. This review should be published, with a several minor issues to be amended during revision.

Page 3: it is known that in addition to astrocytes, fibroblasts are also major components of the glial responses in the lesion site. Such fibrotic scar could be very harmful for tissue dysfunction. The authors should modify the text to reflect this. 

Page 6: There are quite a number of important studies about the roles of potassium transporters in regulating SCI-associated spasticity (for example: Boulenguez, P., et al. (2010). Down-regulation of the potassium-chloride cotransporter KCC2 contributes to spasticity after spinal cord injury. Nat. Med. 16, 302–307) and excitability regulation/functional recovery after SCI (for example Chen et al.,  Reactivation of Dormant Relay Pathways in Injured Spinal Cord by KCC2 Manipulations. 174, 521, 2018). These results should be mentioned.

As mentioned above, the authors should point out studying the detailed cation dynamics as an important future direction. This might be more feasible, in light of recent development of biosensors for detecting individual cations.

Author Response

Dear Reviewer,
Thank you for your appreciation of our manuscript and the constructive comments and advice. Point-by-point responses to your observations can be found below:
-    Page 3: it is known that in addition to astrocytes, fibroblasts are also major components of the glial responses in the lesion site. Such fibrotic scar could be very harmful for tissue dysfunction – we added new paragraphs regarding the indicated completion of our article in this respect. Thank you very much for the suggestion.
-    Page 6: There are quite a number of important studies about the roles of potassium transporters in regulating SCI-associated spasticity (for example: Boulenguez, P., et al. (2010). Down-regulation of the potassium-chloride cotransporter KCC2 contributes to spasticity after spinal cord injury. Nat. Med. 16, 302–307) and excitability regulation/functional recovery after SCI (for example Chen et al.,  Reactivation of Dormant Relay Pathways in Injured Spinal Cord by KCC2 Manipulations. 174, 521, 2018). These results were mentioned in our revised version. Thank you very much for your advice.
-    The detailed cation dynamics research was emphasized as an important future direction. Indeed, this might be more feasible in light of the recent development of biosensors for detecting individual cations. Thank you very much for the suggestion.

Round 2

Reviewer 1 Report

In this second round of revision, the authors updated the main text of the systematic review, updated figure 1, and added a new figure. With these changes, the manuscript showed an increase in presentation quality, but there are still a few issues to point out:

Bellow the major issues:

1)   Figure 2: The text described within the figure can be used as part of the figure caption. The quality of the figure does not allow us to see all the information shown in the left chart. Additionally, the left chart is very complex, and a more detailed explanation is required. An overall increase in the Figures’ appearance and formatting is suggested.

2)   On page 2, lines 72 -77, there are duplicated sentences, and Table 1 is also duplicated there.

3)   On page 3, lines 83 – 89, some sentences are the exact copy as published in the reference paper (doi: 10.3390/cells11040721). Even though the paper is referenced, the exact transcription of a published text is considered a type of plagiarism, so this matter must be urgently addressed.

4)   The same as described above, happened again on Page 4, lines 140 – 143 (https://doi.org/10.3390/cells10112995), and lines 143 – 146 (https://doi.org/10.1152/physrev.00017.2017)

Author Response

Dear reviewer, thank you for this second round of revision. Regarding the issues pointed out, below are our answers:

1)   Figure 2 was significantly improved, respecting your very constructive suggestions for which we wormly thank you! 

2)   On page 2, lines 72 -77, there are duplicated sentences, and Table 1 is also duplicated there. - this situation was generated from manuscript handling, but resolved after acceptance of changes from the previous version.

3)   On page 3, lines 83 – 89, some sentences are the exact copy as published in the reference paper (doi: 10.3390/cells11040721) - the paragraph was modified in order to reduce similarity.  

4)   The same as described above, happened again on Page 4, lines 140 – 143 (https://doi.org/10.3390/cells10112995), and lines 143 – 146 (https://doi.org/10.1152/physrev.00017.2017)  - the paragraph containing the mentioned lines was modified in order to avoid similarity.

Reviewer 2 Report

This revised manuscript has been edited extensively and the submission is now much stronger 

Author Response

Dear reviewer, thank you very much for this second round of revision.

Reviewer 3 Report

This revised manuscript is suitable for publication.

Author Response

Dear reviewer, thank you very much for this second round of revision!